# On transversality and the characterization of finite time hyperbolic subspaces in chaotic attractors

Terence J. O'Kane<sup>1,2</sup> and Courtney R. Quinn<sup>2,1</sup>

<sup>1</sup>CSIRO Oceans and Atmosphere, Battery Point, Hobart, Tasmania, Australia

**Correspondence:** Terence J. O'Kane (terence.okane@csiro.au)

Abstract. We examine the local stable and unstable manifolds of chaotic attractors and their associated growth rates for the quantification of (non-)hyperbolicity in low dimensional nonlinear autonomous dissipative models. This is motivated by a desire for a deeper understanding of transversality and hyperbolicity to inform key challenges to prediction in spatially extended chaotic systems in geophysical flows. In particular, we apply local measures of chaos to quantify the relationship between transversality, dimension, and hyperbolicity on the subspaces of the attractors' invariant manifolds. We consider unstable directions and growth rates determined over finite time intervals, specifically those predicated on information over the past evolution i.e., finite time backwards Lyapunov vectors, and those that include information from both the past and future i.e., finite time covariant Lyapunov vectors. Our study reveals general properties across a diverse set of chaotic attractors at short, intermediate and extended forecast horizons associated with the emergence of distinct locally evolving regions of instability.

#### 1 Introduction

Lorenz (1963) famously introduced his three-variable nonlinear autonomous dissipative model as a simplification of the Saltzman (1962) nonperiodic model of convection. The now famous *L63* model is but one of a number of low dimensional attractors, some also derived by Lorenz himself (Lorenz, 1993), that over the decades have transformed the mathematical study of chaotic systems. These simple sets of coupled ordinary differential equations describing complex trajectories through phase space provide deep insight into many physical phenomena, and in particular the atmosphere - the primary inspiration for Lorenz's exploration. Motivated by the perspectives questions posed by Ginelli et al. (2007), our current investigation applies a hierarchical decomposition of various chaotic attractors. This approach provides a deeper understanding of predictability in nonlinear models via knowledge of the local transversality of the invariant manifolds in combination with information on the past evolution of the unstable phase space trajectories. Specifically, we are interested in how directions of contraction and expansion in phase space (hyperbolicity) and the angles between them (transversality) vary according to chosen temporal window lengths, inform on and characterize the local predictability of the flow.

Lorenz (1965) made a pioneering study of predictability in weather prediction considering the growth of small errors in a low order atmospheric model showing how these were related to the singular values of the tangent linear propagator. Singular vec-

<sup>&</sup>lt;sup>2</sup>University of Tasmania, Sandy Bay, Hobart, Tasmania, Australia

tors (SVs) were subsequently employed in operational numerical forecasting centers implemented as empirically determined combinations of finite-time right (initial) and left (evolved) SVs (Leutbecher and Palmer, 2008). Frederiksen (1997, 2000) had earlier proposed finite-time normal modes (FTNMs) of the propagator as norm independent ensemble perturbations in predictability studies of atmospheric blocking. In particular, Frederiksen (2023) examines the relationships between covariant Lyapunov vectors (CLVs), orthonormal Lyapunov vectors (OLVs), Floquet vectors, finite-time normal modes (FTNMs) and SVs in aperiodic systems. He established asymptotic convergence demonstrating that in the long-time limit, when SVs approach OLVs, the Oseledec theorem and the relationships between OLVs and CLVs can be used to connect CLVs to FTNMs in this phase-space. He documents the conditions on the dynamical systems required to establish convergence to the FTNMs, in terms of ergodicity and boundedness where the FTNM characteristic matrix and propagator is nonsingular. For additional comprehensive reviews of that development, including applications to ensemble prediction, see Buizza et al. (1993); Molteni et al. (1996); Kalnay (2003); Quinn et al. (2021); Frederiksen (2023).

For dissipative chaotic systems i.e., those with at least one positive Lyapunov exponent whose trajectories are bounded within a hyperbox, and whose attractor occupies zero volume in phase space having non-integer dimension less than the number of independent variables of the governing system of equations, the initial evolution is governed by linear dynamics,  $e^{\lambda_j t}$  expanding in the direction(s) where the Lyapunov exponents  $\lambda_j > 0$  and contracting where  $\lambda_j 

chaotic attractor, Quinn et al. (2020) applied a measure of the local attractor dimension in terms of a finite-time Kaplan—Yorke dimension (dimKY) to prescribe the time-dependent rank of the background covariance matrix constructed by projection onto FTCLVs. This measure was constructed via a variable number of weighted finite time covariant Lyapunov exponents where the alignments of the associated FTCLVs were shown to be key to understanding diverse dynamics of disparate regions of the chaotic attractor despite having very similar and even nearly identical local dimension. They specifically investigated the ability to track the nonlinear trajectory in each of the respective subsystems of the 9-component "ENSO coupled with an extratropical atmosphere" of Peña and Kalnay (2004). They showed that, in order to accurately track the trajectory, simply spanning the subspaces of the respective global unstable and neutral modes is not sufficient at times where the nonlinear dynamics and intermittent linear error growth along a stable direction combine. This is due to the fact that the unstable subspace is a function of the underlying trajectory and hence locally defined (Bocquet and Carrassi, 2017). Using observed weather variables Fraedrich (1986) estimated a dimensional value of between three and six for synoptic atmospheric flows and predictability up to 14 days. This approximate range was given further support by the subsequent study of Essex et al. (1987). Using machine learning methods, Axelsen et al. (2025) derived reduced order chaotic models of coherent synoptic atmospheric flows in the Southern Hemisphere of similar dimensionality to those reported by Fraedrich (1986) with lifecycles of ≈ 10 days.

For context, as our motivation is to better understand geophysical dynamical systems, these are typically not hyperbolic (i.e., stable and unstable manifolds are not everywhere transversal), but characterized by the local expanding or contracting directions of a set of leading physical modes. CLVs can be defined from the intersection of the subspaces spanned by tangent linear FTBLEs and their adjoint the FT-*forward*-LEs (FTFLEs) (Vannitsem, 2017) hence growing in time at the rate and directions given by the local Lyapunov vectors (Kalnay, 2003). Importantly, CLVs localized in physical space, provide an intrinsic, hierarchical decomposition of spatiotemporal chaos (Trevisan and Pancotti, 1998; Ginelli et al., 2007) with diverse applications from the formation and persistence of metastable synoptic weather systems (Axelsen et al., 2025) to chaos in semiconductor lasers (Beims and Gallas, 2016).

Based on the aforementioned explorations, we are interested here to characterize how predictability in specific regions of phase space vary with the time widow for evolution in low dimensional chaotic attractors consisting of between 3 and 9 ODEs. The methods we are employing to calculate FTBLEs, FTCLEs and FTCLVs allow for identification of various unstable subregions through a detailed analysis of growth rates, transversality, hyperbolicity and dimension however, at the cost of restricting our analysis to linear error growth (Nese, 1989; B. Eckhardt and Yao, 1993; Ziehmann et al., 2000).

## 2 Method

Ruelle (1979b) first described Oseledec splitting for invertible dynamics as the local decomposition of coordinate independent phase space into covariant directions of the Lyapunov vectors. Ginelli et al. (2007) introduced an algorithm to determine the set of points in phase space whose directions are invariant under time reversal and covariant with the dynamics arguing that these CLVs are coincident with the Oseledec splitting for any invertible dynamical system. A dynamical system is said to be hyperbolic if its phase space has no homoclinic tangencies; i.e., the stable and unstable manifolds are everywhere transversal

100

125

to each other and that this is connected to hyperbolicity (Bochi and Viana, 2004). The determination of the angle between any given pair of CLVs allows for testing the degree of hyperbolicity at any point on the attractor where increasingly larger alignments indicates decreasing degrees of hyperbolicity and visa-versa. Many methods for calculating Lyapunov exponents are available, including recent machine learning approaches (Ayers et al., 2023). Here we use a QR decomposition to calculate the finite time backwards Lyapunov exponents (FTBLEs) (Dieci et al., 1997; Van Vleck, 2010; Dieci et al., 2011). The computation of FTLEs over a finite window of time allows a time-dependent measure of the local unstable, neutral, and stable exponents of the evolving system which approach their asymptotic values as the window length increases.

Of interest here are the local dynamics of the respective chaotic attractors as measured in terms of their finite-time growth rates, hyperbolic splitting on the attractor tangent space (local manifold) measured in terms of alignment of the associated local Lyapunov vectors and dimensionality via the local Kaplan-Yorke dimension. The FTBLEs represent forward evolution over the past period defined by the chosen time window hence directly informing on how predictability varies on the attractor (Arbanel et al., 1991; Yoden and Nomura, 1993). Applying the QR decomposition over finite time windows optimizes mixed initial and evolved singular vectors such that they are no longer infinitesimal but are also of finite size where the chosen window enables exploration of the attractors multiscale nature. Of primary interest here is the application of methods for calculating covariant Lyapunov vectors to measure the degree of hyperbolicity in the local dynamics of the chaotic system. Quinn et al. (2020) showed that very different degrees of hyperbolicity can be manifest at times where nearly identical values of the local Kaplan-Yorke dimension occur, and that the local dimension is insufficient to characterize the finite time dynamics of the particular subspaces occuring on chaotic attractors for given temporal windows. Axelsen et al. (2025) introduced an average measure of hyperbolicity in terms of the mean alignment of FTCLVs at any given point in time however, here we are interested in local hyperbolic subspaces on the attractor and so we calculate these metrics at each point on the phase space trajectory.

As CLVs only truly exist in the asymptotic limit, FTCLVs are more correctly described as mixed initial and evolved singular vectors over some time window given a set of tangent linear propagators. Specifically, Oseledets (1968) theorem relates the Lyapunov exponents  $\lambda_i$  and a non-unique set of vectors  $\phi$  via the forward and backward mapping of the tangent dynamics (cocycle)  $\mathcal{A}(x(t), \tau)$  as

$$\lambda_i = \lim_{\tau \to \infty} \frac{1}{\tau} \log \|\mathcal{A}(x(t), \tau)\phi\| \quad \Longleftrightarrow \, \phi \in \frac{\Phi_i(x(t))}{\Phi_{i+1}(x(t))}$$

For the systems considered here,  $\mathcal{A}(x(t),\tau) = \exp^{\tau \mathcal{J} f(x(t))}$  where  $\mathcal{J}$  is the Jacobian of the right-hand side of any given systems of ODEs considered. For any given CLV pair, we define their alignment as  $\theta_{(i,j)\in\mathcal{N}} = |\phi_i \cdot \phi_j|/(\|\phi_i\| \cdot \|\phi_j\|)$ . Correspondingly,  $\theta_{i,j} = \|\cos(\Theta_{i,j})\|$  given  $\Theta_{i,j}$  is the angle between the  $i^{th}$  &  $j^{th}$  CLV, hence alignment is bounded between [0,1]. Correspondingly for  $\theta_{i,j} = 0$  the CLVs are orthogonal, and for  $\theta_{i,j} = 1$  completely aligned.

To calculate the CLVs we employ Algorithm 2.2 of Froyland et al. (2013) described in table 1. Following the algorithm, matrix cocycles are constructed and a singular value decomposition performed on each, after which the left singular vectors are sorted in descending order based on their singular values. The algorithm then performs a push forward operation over a defined window using the cocycle matrices then reorthogonalizing and repeating until we have a complete set of FTCLVs at a given point in time. For simplicity, we have used a common window  $\delta t$  for calculating the FTBLEs (window); FTCLEs ( $M_GR$ ); and

for the push forward cocyle window (M) used for calculating the CLVs i.e.,  $\delta t = window = M_G R = M$ . For a more detailed discussion of the numerical algorithm see Froyland et al. (2013) and Appendix B of Axelsen et al. (2025). Throughout we use an orthogonalization step of 1.

**Table 1.** Algorithm 2.2 from Froyland et al. (2013) - Approximate the set of N CLVs at time  $t^j$ 

- Construct tangent linear propagators (matrix cocycles)  $\mathcal{A}(x^{i+m},0)$  for every  $m \in [-N,\ldots,N]$
- $\bullet \text{ Compute the eigenvectors } e^{i-N}_j \text{ of } \mathcal{A}(x^{i-N},N)^* \mathcal{A}(x^{i-N},N) \text{ where } \mathcal{A}(x^{i-N},N) = \mathcal{A}(x^i,0) \cdot \ldots \cdot \mathcal{A}(x^{i-N},0)$
- \* denotes adjoint.
- Push forward by multiplication of matrix cocycle,  $\phi_j^i = \mathcal{A}(x^{i-N}, N)e_j^{i-N}$ .
- For each j, reorthogonalize  $\phi^i_j$  with subspace spanned by eigenvectors  $e^{i-n}_k$  for  $k=1,\ldots,j-1$  of  $\mathcal{A}(x^{i-n},N)^*\mathcal{A}(x^{i-n},N)$  every n time steps.
- $\phi_j^i$  approximates the  $j^{th}$  largest CLV at time  $t t_i$ .

We ascertain an approximation to the local attractor dimension based on either the FTBLEs or FTCLEs via the Kaplan-Yorke dimension (Frederickson et al., 1983; Kaplan and Yorke, 2006)

$$dim_{KY} := j + \frac{\sum_{i=1}^{j} \lambda_i}{|\lambda_{j+1}|},\tag{1}$$

where j is the largest leading eigenvector such that  $\sum_{i=1}^{j} \lambda_i \geq 0$  and  $\sum_{i=1}^{j+1} \lambda_i 

| Attractor | ODEs                                                                                                               | parameters                                                                                    | initial conditions                       | timestep             |
|-----------|--------------------------------------------------------------------------------------------------------------------|-----------------------------------------------------------------------------------------------|------------------------------------------|----------------------|
| L63       | Lorenz (1963)                                                                                                      |                                                                                               |                                          |                      |
|           | $\dot{x} = \sigma(y - x)$ $\dot{y} = \rho x - y - xz$ $\dot{z} = xy - \beta z$                                     | $\rho = 28.0$ $\sigma = 10.0$ $\beta = 8.0/3.0$                                               | x(0) = 5.0<br>y(0) = 1.0<br>z(0) = 5.0   | $\triangle t = 0.01$ |
| Dradas    | Ahmad et al. (2024)                                                                                                |                                                                                               |                                          |                      |
|           | $\dot{x} = y - \alpha x + \beta yz$ $\dot{y} = \gamma y + z(1 - x)$ $\dot{z} = \delta xy - \epsilon z$             | $\alpha = 3.0$ $\beta = 2.7$ $\gamma = 1.7$ $\delta = 2.0$ $\epsilon = 9.0$                   | x(0) = -5.0 $y(0) = -5.0$ $z(0) = -15.0$ | $\triangle t = 0.05$ |
| Fourwing  | Qi et al. (2009)                                                                                                   |                                                                                               |                                          |                      |
|           | $\dot{x} = \alpha x + \gamma y z$ $\dot{y} = x(\beta - z) + \delta y$ $\dot{z} = \epsilon z + \rho x y$            | $\alpha = 0.2$ $\beta = 4.0$ $\gamma = 8.0$ $\delta = 1.0$                                    | x(0) = 0.1<br>y(0) = 0.1<br>z(0) = 0.1   | $\triangle t = 0.01$ |
| Hadley    | Sprott (2003)                                                                                                      |                                                                                               |                                          |                      |
|           | $\dot{x} = -y^2 - z^2 - \alpha(x - \gamma)$ $\dot{y} = xy - \beta xz - y + \delta$ $\dot{z} = \beta xy + z(x - 1)$ | $\alpha = 0.2$ $\beta = -0.01$ $\gamma = 1.0$ $\delta = -0.4$ $\epsilon = -1.0$ $\rho = -1.0$ | x(0) = 1.0<br>y(0) = 1.0<br>z(0) = 1.0   | $\triangle t = 0.01$ |

**Table 2.** Attractor definitions used in all subsequent figures and analyses.

| Attractor      | ODEs                                                                                                                                                                                                                                                                                                                                                                                                                                                                    | parameters                                                                                                                                   | initial conditions                                                                                                                        | timestep              |
|----------------|-------------------------------------------------------------------------------------------------------------------------------------------------------------------------------------------------------------------------------------------------------------------------------------------------------------------------------------------------------------------------------------------------------------------------------------------------------------------------|----------------------------------------------------------------------------------------------------------------------------------------------|-------------------------------------------------------------------------------------------------------------------------------------------|-----------------------|
| Threescroll    | Li (2008)                                                                                                                                                                                                                                                                                                                                                                                                                                                               |                                                                                                                                              |                                                                                                                                           |                       |
|                | $\dot{x} = \alpha(y - x) + \delta xz$ $\dot{y} = \beta(x - z) + \rho y$ $\dot{z} = \gamma z + x(y - \epsilon x)$                                                                                                                                                                                                                                                                                                                                                        | $\alpha = 32.48$ $\beta = 45.84$ $\gamma = 1.18$ $\delta = 0.13$ $\epsilon = 0.57$ $\rho = 14.7$                                             | x(0) = 0.1<br>y(0) = 1.0<br>z(0) = -0.1                                                                                                   | $\triangle t = 0.001$ |
| Caputo         | Yan et al. (2022)                                                                                                                                                                                                                                                                                                                                                                                                                                                       |                                                                                                                                              |                                                                                                                                           |                       |
|                | $\dot{x} = \alpha y$ $\dot{y} = -\beta x + \gamma y z$ $\dot{z} = \rho - \delta y^2 + u^2$ $\dot{w} = \epsilon y^2 - w$ $\dot{u} = z$                                                                                                                                                                                                                                                                                                                                   | $\alpha = 3.0$ $\beta = 8.0$ $\gamma = 7.0$ $\delta = 6.0$ $\epsilon = 2.0$ $\rho = 3.0$                                                     | x(0) = 1.0 $y(0) = 0.0$ $z(0) = 0.0$ $w(0) = 0.0$ $u(0) = 0.0$                                                                            | $\triangle t = 0.01$  |
| Penakalnay2004 | Peña and Kalnay (2004)                                                                                                                                                                                                                                                                                                                                                                                                                                                  |                                                                                                                                              |                                                                                                                                           |                       |
|                | $\dot{x_e} = \sigma(y_e - x_e) - c_e(Sx_t - k_1)$ $\dot{y_e} = \rho x_e - y_e - x_e z_e + c_e(Sy_t + k_1)$ $\dot{z_e} = x_e y_e - \beta z_e$ $\dot{x_t} = \sigma(y_t - x_t) - c(SX + k_2) - c_e(Sx_e + k_1)$ $\dot{y_t} = \rho x_t - y_t - x_t z_t + c(SY + k_2) + c_e(Sy_e + k_1)$ $\dot{z_t} = x_t y_t - \beta z_t + c_z Z$ $\dot{X} = \tau \sigma(Y - X) - c(x_t + k_2)$ $\dot{Y} = \tau(\rho X - Y - SXZ) + c(y_t + k_2)$ $\dot{Z} = \tau(SXY - \beta Z) - c_z z_t$ | $\rho = 28.0$ $\sigma = 10.0$ $\beta = 8.0/3.0$ $S = 1.0$ $\tau = 0.1$ $c = 1.0$ $c_{c} = 1.0$ $c_{e} = 0.08$ $k_{1} = 10.0$ $k_{2} = -11.0$ | $x_e(0) = -5.0$ $y_e(0) = -5.0$ $z_e(0) = 15.0$ $x_t(0) = -5.0$ $y_t(0) = -5.0$ $z_t(0) = 15.0$ $X(0) = -5.0$ $Y(0) = -5.0$ $Z(0) = 15.0$ | $\triangle t = 0.005$ |

**Table 3.** Attractor definitions used in all subsequent figures and analyses.

140

145

155

165

Figure 1 shows FTBLEs and corresponding instantaneous  $dim_{KY}$  values for the Lorenz 'butterfly attractor' (Lorenz (1963): L63) for windows  $\delta t = 5, 25, 50, 100, 200$ . Figure 2 shows FTCLEs and corresponding  $dim_{KY}$  and in figure 3 we depict the alignment  $\theta_{i,j}$  between the FTCLV pairs for each of the five chosen windows. The differences between FTBLEs and FTCLEs are immediately apparent most notably in the second exponent. In general, it is noticeable that where FTBLE1 is unstable, FTBLE2 is largely stable. FTBLE3 is always stable with the largest absolute values occurring where FTBLE1 is unstable. For cocyle windows  $\delta t = 5, 25, 50$ , FTBLE1 is largely unstable in the region of the saddle and on a restricted region of the inner orbits of each wing of the attractor. As the window is increased to  $\delta t = 100$ , unstable values are compressed to regions near the saddle and between the fast and slow orbits of the attractor wings. For windows  $\delta t > 100$  FTBLE2 assumes larger unstable values on the inner and outer loops. As window length increases the FTBLE3 values become increasingly less stable. The figure  $1 \ dim_{KY}$  plots reflect the combined contributions of the FTBLEs to the attractor dimension. As forecast window increases the stable subregions evident at  $\delta t = 5, 25, \&50$  shrink where upon for  $\delta t > 100$  the attractor is essentially unstable everywhere as expected. As  $\delta t \to \infty$ ,  $dim_{KY}$  is seen to approach its asymptotic value at all points on the attractor.

The growth rate of FTCLE1 mirrors that of FTBLE1, however the stable subregions evident for  $\delta t=5$  are considerably reduced in comparison. Further, we see (figure 2) FTCLE2 is stable and increasingly so in the outer loops as  $\delta t \to 100$ . However, at  $\delta t=200$  the extent of the most stable regions of FTCLE2 reduces by  $\approx 55\%$ . The FTCLE-based  $dim_{KY}$  largely reflects the subregion structure of FTCLE1 values on the attractor. In general, the mean values of the FTBLEs do not change appreciably however, those for the FTCLEs are highly variable. Considering  $\theta_{i,j}$  (figure 3) we see the region of very low dimension evident in  $dim_{KY}$  for  $\delta t=5$  (figure 2) corresponds very closely to the highly localized region of alignment evident between  $\theta_{1,2}$ , otherwise there is minimal to no alignment elsewhere on the attractor. At  $\delta t=25$  alignment near the same region becomes very low forming a locally hyperbolic subregion in addition to one near the saddle. As the window  $\delta t>25$  increases,  $\theta_{1,2}$  alignment becomes ubiquitous in all regions away from the saddle. For  $\delta t=5$ ,  $\theta_{1,3}$  and  $\theta_{2,3}$  exhibit values >0.5 only on the same two subregions of the outer orbits of the attractor. For  $\delta t=100,200$ ,  $\theta_{1,3}$  and  $\theta_{2,3}$  values <=0.5 correspond to subregions on the attractor where  $dim_{KY}>2.0$  (figure 2). Hence at  $\delta t=200$  it appears that regions with large FTCLE1 values i.e., >0.7, are permissible due to the correspondingly low alignments  $\theta_{1,3}$  and  $\theta_{2,3}$  compensating the high alignments  $\theta_{1,2}$ .

Next we consider the three wing Dradas attractor. Dradas FTBLEs & FTCLEs are shown in figures 4 & 5 for  $\delta t = 5,50,\&400$  respectively. Both FTBLEs & FTCLE growth rates show very similar subregions for each of the considered values of  $\delta t$ . For  $\delta t = 5$  FTBLE1 & FTCLE1 two distinct unstable subregions are visible on two of the attractor wings while the third is everywhere stable. FTBLE2 & FTCLE2 have similar corresponding regional structures although the unstable FTCLE2 subregions are more restricted relative to FTBLE2. FTBLE3 & FTCLE3 are stable everywhere on the attractor with mean values many times larger than that of the leading exponent signifying a highly extended system. At  $\delta t = 50$  the values of the leading exponent becomes unstable on the inner orbits of the attractor as those of the second exponent become stable. As  $\delta \to \infty$  all FTBLE & FTCLE values at any given point on the attractor approach their mean asymptotic value. While the mean FTBLE values are relatively unchanged as  $\delta t \to \infty$ , the absolute values of FTCLEs2 & 3 reduce as they become increasingly less stable and the system less extended. Despite this the  $dim_{KY}$  values (figure 5) on the attractor are very similar regardless of being calculated using FTBLEs or FTCLEs. The Dradas alignments (figure 6) are substantially more complicated and

Figure 1. L63: FTBLEs 1, 2, & 3 values at each point on the attractor in x-y-z orientation for windows  $\delta t = 5, 25, 50, 100, 200$ . The far right column displays corresponding  $dim_{KY}$  values based on the instantaneous FTBLE values.

Figure 2. L63: As for figure 1 but for FTCLEs.

Figure 3. L63:  $\theta_{i,j}$  pairs in x-y-z orientation. Values of 1 and 0 respectively indicate complete alignment or exact orthogonality.

Figure 4. Dradas: FTBLEs 1, 2, & 3 values at each point on the attractor in x-y-z orientation for windows  $\delta t = 5, 50, 400$ . The far right column displays corresponding  $dim_{KY}$  values based on the instantaneous FTBLE values.

less easily interpreted with respect to those observed for L63. However, for  $\delta t=5$  we can recognize regions where all three FTCLVs are aligned such as the lower wing of the attractor, corresponding to stable subregions on the attractor with  $dim_{KY}$  values approaching zero. At  $\delta t=50$  we see these subregions contract to distinct bands on the lobes after which the alignments  $\theta_{1,3}$  and  $\theta_{2,3}$  respectively break down becoming diffuse and unstructured at  $\delta t=400$ .

The fourwing (Qi et al., 2009; Wang et al., 2010) and Hadley (Sprott, 2003) attractors for  $\delta t=5$  are both hyperbolic at all points on the attractor with no stable subregions evident i.e.,  $dim_{KY}>0$  everywhere (figures 7 & 8). Both attractors show distinct FTCLE2 subregions of either growth or decay whereas those of the leading FTCLE1 are everywhere unstable and for the FTCLE3 everywhere stable. At  $\delta t=100$  fourwing  $\theta_{1,3}$  amd  $\theta_{2,3}$  alignments occur in the same localized outer regions of the attractor wings with the largest alignment values for  $\theta_{1,2}$  (figure 7). Fourwing  $dim_{KY}$  values resemble those of FTCLE2 being largest where FTCLE1&2 are coincidentally unstable and smallest where FTCLE2&3 are stable. Similar relationships

Figure 5. Dradas: As for figure 4 but for FTCLEs.

between the growth rates and vector alignments occur for the Hadley attractor (figure 8) with one noticeable difference. For  $\delta t=100$  we see the leading FTCLE1 indicate distinct regions of contraction and  $\theta_{i,j}$  values correspondingly indicative of significant alignment between all vectors. In this case  $dim_{KY}>0.5$  occur over a very restricted region where FTCLE1 growth rates are >=0.75. FTCLE2 becomes everywhere stable with mean value approaching that of FTCLE3 hence determining the generally low  $dim_{KY}$  values.

The threescroll attractor (figure 9) exhibits similar characteristics to those of Hadley and fourwing. At  $\delta t=5$  the system exhibits low alignment values everywhere with nearly uniform growth rates at points on the attractor. FTCLEs1 & 2 are everywhere unstable and FTCLE3 stable. This is reflected in  $dim_{KY}$  at points on the attractor are close to the average  $dim_{KY}\approx 2.5$ . At  $\delta t=50$  mean values indicate contraction on most of the attractor as the ratio of  $FTCLE1/FTCLE3\approx 0.74$  at  $\delta t=5$  changes significantly to  $FTCLE1/FTCLE3\approx 0.34$  as  $\delta t\to 50$ . Hence the system becomes more extended with regions of

**Figure 6.** Dradas:  $\theta_{i,j}$  pair values in x-y-z orientation with elevation angle 30° and azimuthal angle 0°. Values of 1 and 0 respectively indicate complete alignment or exact orthogonality.

205

Table 4. Attractor FTBLEs and growth rates

|            | L63     |          | Dradas   |         | Fourwing |         | Hadley  |         |
|------------|---------|----------|----------|---------|----------|---------|---------|---------|
| $\delta t$ | FTBLE   | FTCLE    | FTBLE    | FTCLE   | FTBLE    | FTCLE   | FTBLE   | FTCLE   |
|            | 0.9024  | 4.0870   | 0.5532   | 1.2021  | 0.0680   | 0.2985  | 0.0016  | 2.2922  |
| 5          | -0.0032 | -2.2811  | -1.4719  | -1.9427 | -0.0014  | -0.0872 | -0.0225 | 0.0556  |
|            | -0.1457 | -15.6224 | -10.5044 | -8.6167 | -1.2694  | -1.4005 | -0.0227 | -2.4052 |
|            | 0.9022  | 2.7985   |          |         |          |         |         |         |
| 25         | -0.0023 | -2.4165  |          |         |          |         |         |         |
|            | -0.1457 | -14.8132 |          |         |          |         |         |         |
|            | 0.9015  | 1.7964   | 0.5532   | 0.9469  |          |         | 0.0013  | 1.1120  |
| 50         | -0.0022 | -2.3656  | -1.4721  | -1.3270 |          |         | -0.0225 | -0.7318 |
|            | -0.1457 | -14.5736 | -10.5044 | -7.9628 |          |         | -0.0273 | -2.0670 |
|            | 0.8998  | 0.8028   |          |         | 0.0676   | 0.2226  | 0.0010  | 0.0269  |
| 100        | -0.0013 | -1.7310  |          |         | 0.0014   | -0.1270 | -0.0230 | -1.2879 |
|            | -0.1465 | -14.3193 |          |         | -1.2689  | -1.3837 | -0.0267 | -1.6414 |
|            | 0.8964  | 0.8870   |          |         |          |         |         |         |
| 200        | -0.0050 | -0.9483  |          |         |          |         |         |         |
|            | -0.1456 | -14.1702 |          |         |          |         |         |         |
|            |         |          | 0.5538   | 0.9456  |          |         |         |         |
| 400        |         |          | -1.4718  | -0.7541 |          |         |         |         |
|            |         |          | -10.5047 | -0.7756 |          |         |         |         |

high  $dim_{KY}$  occurring where alignments  $\theta_{1,2}$ , and to a lesser extent  $\theta_{1,3}$ , are 

Figure 7. Fourwing:  $\theta_{i,j}$  values in x-y-z orientation with elevation angle  $30^{\circ}$  and azimuthal angle  $0^{\circ}$ . FTCLEs 1, 2, & 3 and  $dim_{KY}$  for  $\delta t = 5{,}100$ .

**Figure 8.** Hadley: FTCLEs,  $\theta_{i,j}$  and  $dim_{KY}$  on the three dimensional projection of the attractor shown at  $\delta t = 5,100$ .

Figure 9. Threescroll: FTCLEs and  $dim_{KY}$  on the three dimensional projection of the attractor;  $\theta_{i,j}$  pairs on chosen axes; shown at  $\delta t = 5,50$ .

Figure 10. Caputo:  $dim_{KY}$  for given  $\theta_{i,j}$  pairs on chosen axes.

**Table 5.** Attractor FTBLEs and growth rates. Bracketed values indicate approximate asymptotic backwards Lyaponov Exponent (LE) values  $(\delta t = 400)$  previously reported by Quinn et al. (2020)

|            | Threescroll |          | Caputo  |         | Penakalnay2004       |          |
|------------|-------------|----------|---------|---------|----------------------|----------|
| $\delta t$ | FTBLE       | FTCLE    | FTBLE   | FTCLE   | FTBLE                | FTCLE    |
|            |             |          |         |         | 0.9059               | 5.0825   |
|            |             |          |         |         | 0.2848               | 2.2879   |
|            |             |          | 0.01876 | 3.7769  | -0.0001              | 0.7411   |
|            | 0.0607      | 34.0406  | 0.0022  | 0.1727  | -0.0093              | -0.4965  |
| 5          | -0.0014     | 5.7347   | -0.0033 | -0.4129 | -0.3861              | -1.8623  |
|            | -0.5065     | -45.9772 | -0.0564 | -0.9096 | -0.7855              | -2.4140  |
|            |             |          | -0.9607 | -4.1289 | -1.7706              | -3.3905  |
|            |             |          |         |         | -12.3690             | -12.6146 |
|            |             |          |         |         | -14.5700             | -15.7245 |
|            |             |          |         |         | 0.9075               | 1.1619   |
|            |             |          |         |         | 0.2861               | -0.3507  |
|            |             |          | 0.1741  | 0.1297  | 0.0006               | -0.7403  |
|            | 0.0568      | 5.2889   | 0.0013  | -0.6292 | -0.0088              | -1.0320  |
| 50         | -0.0021     | -7.6030  | -0.0036 | -1.4028 | -0.3856              | -1.9801  |
|            | -0.4931     | -15.5595 | -0.0564 | -1.6979 | -0.7836              | -2.7733  |
|            |             |          | -0.9609 | -2.3770 | -1.7719              | -3.5542  |
|            |             |          |         |         | -12.3722             | -13.2860 |
|            |             |          |         |         | -14.7200             | -15.1305 |
|            |             |          |         |         | 0.9077 (0.9071)      | 0.3869   |
|            |             |          |         |         | $0.2881 \ (0.2670)$  | -0.7620  |
|            |             |          |         |         | 0.0029  (-0.0056)    | -0.6998  |
|            |             |          |         |         | -0.0081  (-0.0060)   | -0.9757  |
| 100        |             |          |         |         | -0.3843  (-0.4326)   | -1.5272  |
|            |             |          |         |         | -0.7826  (-0.7706)   | -2.3517  |
|            |             |          |         |         | -1.7734  (-1.8263)   | -3.3014  |
|            |             |          |         |         | -12.3753  (-12.2691) | -12.7425 |
|            |             |          |         |         | -14.5751  (-14.5640) | -14.5800 |

relative to the FTBLEs is a general property of all exponents as evident from the values in tables 4 & 5. While choosing to use the FTCLEs rather than the FTBLEs does lead to differences in the structure of the local Kaplan–Yorke dimension stable and unstable subregions, these differences are most evident in the relative magnitudes of the leading unstable and most stable exponents, and tend to diminish as  $\delta t \to \infty$  as in the limit they approach the asymptotic LV values. Shown in the upper three rows of figure 11  $dim_{KY}$  values calculated from FTCLEs are projected onto each of the three subsystems of the PenaKalnay2004 model. Here we see regions of high dimension contracting to the region of the saddle node  $(x_e, y_e, z_e)$  and associated regions where alignments are generally small. The corresponding  $dim_{KY}$  values based on the FTBLEs at  $\delta t = 100$  are shown in last

row in figure 11. Differences between FTBLE and FTCLE  $dim_{KY}$  values at  $\delta t = 100$  are largely in terms of the magnitude of the dimension with the most unstable regions occurring in co-located subregions i.e., differences correspond to a constant scale factor.

### 4 Discussion and conclusions

Takeuchi et al. (2011) provide a framework for understanding hyperbolic decoupling of the tangent space into subspaces in high dimensional spatially extended dissipative systems in which the entangled "physical" modes are separated from the rapidly decaying stable modes. For prediction studies one is typically most concerned with the trajectory of the entangled modes on the associated finite-dimensional tangent space of the phase-space dynamics. This slow manifold is often identified in terms of the spectral gap in the eigenvalues. From the geometrical viewpoint, where the system is reducible to the evolution of a few degrees of freedom, it follows that the flow exists in a low-dimensional region of phase space, parametrized by a finite number of degrees of freedom. For geophysical fluids such as the atmosphere, one of the greatest challenges is to identify the emergence of a low-dimensional manifold in the local spatio-temporal dynamics of high dimensional flows. Such slow-fast hydrodynamic systems are paradigmatic examples with deep roots in statistical physics (Kogelbauer and Karlin, 2024).

Motivated by the work of Lorenz (1993) and Fraedrich (1986), as well as the questions posed by Ginelli et al. (2007), we have investigated hyperbolicity via the relationship between fluctuations of the Lyapunov exponents, transversality of their associated dynamical vectors, and dimensionality. We are further motivated by the recent study of mid-latitude persistent events in the Southern Hemisphere mid-troposphere by Axelsen et al. (2025). They employed an aggregated measure of alignment to indicate hyperbolic splitting of reduced local tangent space dynamics occurring at geographic locations where atmospheric blocking is known to preferentially occur (O'Kane et al., 2016). Here we undertook a more detailed examination of the local dynamics of a diverse set of chaotic attractors, some with characteristics broadly applicable to geophysical flows, to ascertain if commonalities exist.

Our general findings are:

- over short widows  $\delta t 

Figure 11. PenaKalnay2004:  $dim_{KY}$  at  $\delta t = 5, 50, 100$  on each of the three component subsystems (extratropical:  $x_e, y_e, z_e$ ); (tropical:  $x_t, y_t, z_t$ ); & (ocean: X, Y, Z).

250

255

260

265

270

largely unchanged whereas others, like threescroll, becoming inceasingly less stable as  $\delta t$  increases. That said, the ratio of the absolute mean values of the most unstable to most stable exponents is most often observed to increase with window length. For the higher dimensional attractors Caputo and Penakalnay2004, very complex alignments are manifest such that transversality between various vectors and exponent growth rates are complicated. In such cases the attractor dimension, which is an aggregated value of the exponents, is a more readily interpretable indicator of regions of (non)-hyperbolicity. The most complicated dynamics are observed to occur over these intermediate time windows.

- over extended widows  $\delta t>=100$  the unstable subregion of the near neutral exponents evident at intermediate and shorter times tend to become stable on most of the attractor such that only the leading exponent determines regions where the unstable subspaces occur. As  $\delta t\to\infty$  the values of a given exponent approach the mean asymptotic value at all points on the attractor and the subspace regions evident over shorter finite time windows merge and disappear. This is most easily seen for Dradas, the attractor with the most rapid convergence of the FTBLEs and FTCLEs to their mean asymptotic LE value.

Ginelli et al. (2007) proposed that access to the local directions of stable and unstable manifolds and the ready characterization and quantification of (non-)hyperbolicity affords a means to better model the spatial structure of the dynamics in extended systems. In particular, they note the key challenges to quantification of local measures of chaos and hierarchical modal decompositions of spatiotemporal chaos as well as the potential applications to prediction in nonlinear models. In recent years these ideas, including knowledge of the local transversality of invariant manifolds, have indeed been combined with linear and nonlinear generalizations of dynamical vectors using information on the past evolution e.g., SVs, FTBLVs, BVs, etc., to initialize optimal forecast perturbations along the relevant unstable directions determining error growth. Our study reveals that, even given the complexities of the local dynamics of low dimensional chaotic attractors associated with the manifestation of diverse unstable subspaces, there are general properties identifiable in terms of the relationship between transversality and local measures of chaos. We also note that the changing local hyperbolic structure can provide additional information about "nearby" (in parameter space) bifurcations potentially providing "early warning" indicators for tipping points, and that this is an area for further investigation.

Code and data availability. https://github.com/oka005/attractor/archive/refs/tags/v1.0.0.tar.gz

*Author contributions.* TJO wrote the first draft and carried out the computations. Both authors contributed to the investigation, code development and revising the manuscript.

Competing interests. None

275 Acknowledgements. TJO was supported by the Australian Commonwealth Scientific and Industrial Organisation (CSIRO). CRQ was supported by Australian Research Council (ARC) DECRA grant No. DE250101025

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
