# Peer review of "On transversality and the characterization of finite time hyperbolic subspaces in chaotic attractors"

_EGUsphere, 2025_

## Referee Comment (RC2)

**Reviewer Comments for Manuscript "On transversality and characterization of finite time hyperbolic subspaces in chaotic attractors"**

This manuscript investigates the local geometric structure of chaotic attractors within low-dimensional dynamical systems, specifically focusing on the transversality between stable and unstable manifolds. The manuscript apply local measure of chaos to quantify the relationship between transversality, dimension and hyperboicity on the subspaces of the attractors' invariant manifolds. While the theoretical framework is intriguing and the numerical analysis of the low-dimensional models is detailed, there are several key issues that need to be addressed before publication.

1. The analysis is primarily conducted on low-dimensional chaotic models (Eg. Lorenz 63 or a 9-variable model). It is well known that low-dimensional models lack the multi-scale interactions and spatial extensiveness characteristic of high-dimensional turbulent systems. The authors should provide a more in-depth discussion on the validity of extrapolating local hyperbolicity and transversality features observed in these simple attractors to high-dimensional systems.

2. The introduction reviews previous work, such as Quinn et al. (2020), regarding the finite-time Kaplan-Yorke dimension (dimKY) and data assimilation. Given the overlap in the author list and the thematic similarity, it is essential to clearly distinguish the contributions of the current manuscript from these prior studies. Does the novelty lie in the introduction of a new analytical perspective or in understanding the underlying dynamical mechanisms? I recommend explicitly stating the innovations of this work in the Introduction or Discussion.

3. The paper would benefit significantly from a stronger connection to physical mechanisms. Specifically, when the system exhibits some properties, such as "non-hyperbolicity", what does these properties correspond to in terms of the physical state or dynamical events within the model?.

4. Should the second column of Figure 1 be labeled as "FTBLE2" ?